# Characterization of Tensile Stress-Dependent Directional Magnetic Incremental Permeability in Iron-Cobalt Magnetic Sheet: Towards Internal Stress Estimation through Non-Destructive Testing

**DOI:** 10.3390/s22166296

**Published:** 2022-08-21

**Authors:** Borel Toutsop, Benjamin Ducharne, Mickael Lallart, Laurent Morel, Pierre Tsafack

**Affiliations:** 1Faculty of Engineering and Technology, University of Buea, Buea P.O. Box 63, Cameroon; 2Univ Lyon, INSA Lyon, Ecole Centrale de Lyon, Université Claude Bernard Lyon 1, CNRS, Ampère, UMR5005, 69622 Villeurbanne, France; 3Univ Lyon, INSA Lyon, LGEF EA682, 69621 Villeurbanne, France; 4ELyTMaX IRL3757, Univ Lyon, INSA Lyon, Centrale Lyon, Université Claude Bernard Lyon 1, Tohoku University, Sendai 980-8577, Japan

**Keywords:** magnetic control, internal stress, local characterization, domain wall bulging, multi-axis magnetization

## Abstract

Iron-Cobalt ferromagnetic alloys are promoted for electrical energy conversion in aeronautic applications, but their high magnetostrictive coefficients may result in undesired behaviors. Internal stresses can be tuned to limit magnetostriction but must be adequately assessed in a non-destructive way during production. For this, directional magnetic incremental permeability is proposed in this work. For academic purposes, internal stresses have been replaced by homogenous external stress, which is easier to control using traction/compression testbench and results in similar effects. Tests have been limited to tensile stress stimuli, the worst-case scenario for magnetic stress observation on positive magnetostriction coefficient materials. Hysteresis cycles have been reconstructed from the incremental permeability measurement for stability and reproducibility of the measured quantities. The directionality of the sensor provides an additional degree of freedom in the magnetic response observation. The study reveals that an angle of π/2 between the DC (*H_surf DC_*) and the AC (*H_surf AC_*) magnetic excitations with a flux density *B_a_* at *H_surf_*
*_DC_* = 10 kA·m^−1^ constitute the ideal experimental situation and the highest correlated parameter to a homogeneous imposed tensile stress. Magnetic incremental permeability is linked to the magnetic domain wall bulging magnetization mechanism; this study thus provides insights for understanding such a mechanism.

## 1. Introduction

The aeronautic domain is undergoing profound changes, visible in the proliferation of electrical comfort equipment, electronic controllers, and navigation aid systems. As the level of required electrical power increases, so do the mass of electrical appliances since the weight/power ratio remains challenging to reduce [1].

In the case of onboard transformers [2], an industrial solution for the increase in power density consists of developing new magnetic materials such as iron-cobalt alloys. High power density gains (up to 10% and more) are, for instance, expected with Fe-27%Co compound. This material exhibits the highest saturation magnetization of all known soft ferromagnetic materials (>2.4 T) [3].

Unfortunately, FeCo alloys’ astonishing saturation magnetization is associated with substantial magnetostrictive effects. These effects result in large deformation and significant undesired acoustic noise [4,5]. There are many ways to reduce magnetostriction, including the development of appropriate textures [4,6]. In [7], B. Nabi et al. claimed that magnetostriction in Fe-27%Co alloy is a consequence of magnetocrystalline anisotropy partly linked to the sheet crystallographic texture and, by developing a Goss texture, much better magnetic properties will be observed including a lower magnetostriction. Tuning internal stresses is another promising solution [8].

Mechanical Internal Stress (MIS) is crucial in iron alloys’ magnetic and mechanical performance. MIS originates from various mechanisms, including plastic deformations, temperature gradients, or microstructural changes. Manufacturing processes including machining, welding, shot peening, heat treatment, and grinding are sources of MIS. 

All MIS estimation methods are indirect. They start with the measurement of a coupled physical quantity and finish with a calculation stage [9]. The hole-drilling method [10] (see Figure 1 for illustration), the contour method [11], the crack compliance method [12], and the stripping method [9] are the most conventional mechanical methods. Together with the chemical processes [9], they constitute the so-called destructive techniques.

Non-destructive Testing (NDT) methods have also been described [13,14,15]. These methods include diffraction (X-ray, Neutron), ultrasounds, or acoustic emission [16]. In the case of conductive and/or ferromagnetic parts, electromagnetic NDT is often indicated. Eddy currents testing [17], magneto-acoustic emission [18,19], electromagnetic acoustic transducer [6], and Magnetic Barkhausen Noise (MBN) [20,21,22] are the most popular ones [23]. These methods offer fast response, low cost, small size, and easy maintenance, but their current industrial developments are limited, and their efficiency is questionable. Very few of them have crossed the threshold between academia and industry. 

Based on MBN measurements, the Stresstech controller (Jyväskylä, Finland) is, however, one of them [9]. A significant problem for this device comes from a quasi-impossibility to distinguish the effect of MIS from other dependent properties (dislocations, grain size, texture, plastic strain, precipitates, phase changes, impurities, etc.). The micromagnetic, multi-parametric, microstructure and stress analysis (3MA) developed by IZFP Fraunhofer institute (Saarbrücken, Germany) is an interesting alternative. 3MA accumulates and combines data from different magnetization signatures and identifies the ultimate magnetic combination of indicators to a given targeted property (hardness, internal stress, yield strength, etc.) [24]. 3MA is pragmatic and efficient but needs time-consuming experimental campaigns and provides non-transposable results. 

All magnetic MIS controllers have been designed on the same principles that use a unidirectional magnetization induced by a powerful electromagnet combined with local surface sensors to observe the magnetic answers [25]. A current research trend of alternative and/or self-made magnetic sensors for magnetic NDT measurements is perceptible in the literature [26,27]. While attractive in terms of sensor reproducibility and stability, these innovations fail to accurately showcase the claim for directional selectivity of the magnetic sensor as they are applied in angular-dependent measurements. In Figure 14 of [21], however, simulation predictions on FeCo alloys show that stress effects on the magnetic response can be more visible when the magnetic excitation and the stress direction are not colinear and that a directional sensor would probably bring significant improvement in the stress observation. 

Among the magnetic methods, just a few can enforce directional measurements. Point probes [28,29] and MBN are one of them [30] but not tested in this study. Instead, we opted for Magnetic Incremental Permeability (MIP, *μ_MIP_*), as the first results over stress dependency were auspicious ([31], Figure 13 in [32]). MIP is based on the domain walls bulging magnetization mechanism (reversible magnetization variations), as observed under low amplitude magnetic excitation variations. This mechanism appears to be very sensitive to MIS [31].

So far, MIP has almost always been observed with pancake (Figure 2, left-hand side) or wound coils preventing access to directional information. Still, this limitation can be overcome using double coils Transmitter/Receptor adjacent TR-probes as depicted on the right-hand side of Figure 2. 

We propose to overpass the directional limitation in the present study by using a miniature U-shape ferrite magnetic core. MIS is impossible to be precisely controlled; thus, it has been replaced by external stress. The effect on the magnetization process is considered independent of the stress origin. Tests were limited to tensile stress, constituting the worst-case scenario for stress observation. Iron-Cobalt alloys have positive magnetostriction coefficients, and tensile stress softens the magnetic behavior. As an already soft material, limited tensile stress effects can be observed in the *B_a_(H_surf_)* classic hysteresis cycles (where *B_a_* is the flux density, and *H_surf_* is the tangential magnetic excitation field). Hysteresis cycles were reconstructed for all MIP measurements providing a stable and reproducible signature. Then, indicators read on these cycles were defined and plotted vs. the homogeneous tensile stress. Pearson factors were calculated and analyzed to establish their correlation level with the external tensile stress. Conclusions were drawn regarding the non-destructive testing objective and the physical properties of the domain wall bulging. 

This manuscript is organized as follows: Section 2 describes the experimental conditions, including sensors and tested specimens. Section 3 gives the experimental results. Section 4 provides analysis, discussions, and conclusions. Among them, the optimal conditions arise for a mechanical tensile stress estimation based on directional MIP.

## 2. Experimental Setup

### 2.1. Description of the Specimens

FeCo Iron-Cobalt laminations were tested in this study. This material has a high yield strength (1000 MPa). All specimens were extracted from the same batch. Strips were cut by Electrical Discharge Machining (EDM) to get tensile-shape type specimens (Figure 3). Two specimens were tested for consistency.

Table 1 gives the detailed composition, physical and mechanical properties of the tested specimens.

### 2.2. Description of the Experimental Setup

The IEC 60404-3 standard details using a single sheet tester for the magnetic characterization of a ferromagnetic lamination [33]. For geometrical reasons, it is impossible to combine a traction bench while respecting all the recommendations. Still, the experimental setup we developed was inspired by this standard. Figure 4 gives pictures and a 2D overview of the test bench. 

A tension–compression machine Shimadzu AGS-X series (Kyoto, Japan) was used for the tensile stress application. All tests were done at room temperature and under constant imposed stress conditions without initial load. The distance between the grips was 100 mm. Once the constant stress was imposed, a minimum of 60 s was waited before starting the acquisition process to avoid any drift issues. The magnetic inductor was made of a U-shaped FeSi 3% electrical steel yoke. The leg size of the yoke was 12 mm × 12 mm, and the inner distance between the legs was 30 mm. A 500 Turns excitation coil was wound around the yoke and supplied by a Kepco BOP 100-4 M (New York, NY, USA) power amplifier for the *B_a_(H_surf_)* characterization and by an RS pro 180 W bench power supply (Corby, UK) for the MIP. The data acquisition and analog signal generation were ensured by the DEWESoft X2 (Trbovlje, Slovenia) data acquisition software associated with a SIRIUS 8 × CAN data acquisition. A noise-shielded radiometric linear Hall probe SS94A from Honeywell (Charlotte, NC, USA) was positioned tangentially to the surface of the tested sample for the surface magnetic field (*H_surf_*) measurement.

### 2.3. Magnetic Sensors

Two sensors dedicated to evaluating the magnetic state of the tested specimens have been used in this study. Figure 5 depicts a 3D overview of both sensing solutions.

The first sensor was an *n* = 50 turns wound coil. The magnetic flux density *B_a_* was obtained by numerical integration (Equation (1)) of the voltage drop *e(t)* during the magnetization cycle:(1)$$B_{a}\left(t\right)=-\frac{1}{n\cdot S}\overset{t}{\underset{0}{\int }}e\left(t\right)dt$$
where *S* is the specimen cross-section. A numerical correction was done to cancel the undesired drift due to the integration process. The second sensor (Figure 6) combines half a toroidal ferrite core (WE-TOF EMI from Würth Elektronik, Künzelsau, Germany), and a 54 turns wound coil. Specific 3D-printed support (see Figure 5 and Figure 6) has been designed to hold the sensor and authorize plane measurements at every Δ*q* = π/18 rad angle step. A Flashforge Dreamer (Jinhua, China) and PLA polymer were used. This support does not influence the magnetic measurements. 

A precision Keysight (Santa Rosa, CA, USA) LCR meter was used to record the impedance *Z* of the second sensor during the experimental phases.

### 2.4. Experimental Process

The experimental campaign was divided into two phases:The first phase was dedicated to evaluating the tensile stress effect on the evolution of the classic *B_a_(H_surf_)* hysteresis cycles. This effect has already been thoroughly described in the literature [28,34], and our objective was to validate the specimens’ conformity;A similar tensile stress sequence was run in the second phase but combined with directional magnetic incremental permeability measurements. For each tensile stress level, a set of ten *Z(H_surf_)* curves were plotted (for different values of angle *q* from 0 to π/2 rad with a Δ*q* = π/18 rad step).

## 3. Experimental Results

### 3.1. B_a_(H_surf_) Hysteresis Cycles

Tensile stress *σ* on Iron-Cobalt ferromagnetic materials is expected to soften the magnetic behavior. In the low-frequency range, softer magnetic behavior means higher permeability but lower coercivity. This behavior is well-known and has already been displayed in the scientific literature (Figure 6 in [21] and Figure 2.34 in [34]).

Experimental results depicted in Figure 7 confirm our expectations. They validate the conformity of our test bench and the tested specimens.

*B_a_(H_surf_)* hysteresis cycles are firmly tensile stress-dependent and could be considered for *σ* observation. Still, the impossibility of using wound coils in an industrial context where large surfaces must be controlled is worth mentioning. This observation encourages users toward other methods, including the directional MIP non-destructive method described below.

### 3.2. Directional Incremental Permeability Z(H_surf_)

#### 3.2.1. Magnetic Incremental Permeability

In general, magnetic permeability *μ* can be described as a measure of the material’s response to an applied magnetic field.

More precisely and according to the German standard DIN1324 (part II: magnetic field for material quantities [35,36]), the Magnetic Incremental Permeability (MIP, *μ_MIP_*) is defined as the slope of inner asymmetric loops (Equation (2), Figure 8). These loops, also called minor cycles, are obtained when the tested material is exposed to the superimposition of two magnetic contributions:A low-frequency (quasi-static), high amplitude magnetic excitation, that provides a bias magnetization;A high-frequency, low amplitude magnetic excitation, allowing the measurement of the relative magnetic incremental permeability $\mu_{MIP_{r}}$ as:(2)$$B_{a}\left(t\right)=-\frac{1}{n\cdot S}\overset{t}{\underset{0}{\int }}e\left(t\right)dt$$

#### 3.2.2. *Z(H_surf_)* Butterfly Loops

The experimental setup described in Section 2.2 measures the impedance of the MIP sensor shown in Figure 5 and Figure 6. Figure 9 consists of the evolution of *IZI*, the sensor impedance modulus vs. the quasi-static magnetic excitation for four levels of tensile stress *σ*. For this test, the sensor is aligned with the tested specimen’s length and the magnetic excitation (see Figure 9 top left-hand corner).

#### 3.2.3. From *Z* to the FeCo *μ_MIP_*

The interpretation of measurements displayed in Section 3.2.2 is facilitated by returning the physical quantities. For this, the relation between *IZI* and *μ_MIP_* for the FeCo sheet MIP needs to be established. 

The solution opted out in this study is based on a magnetic reluctance scheme. Even inexact as relying on simplifying assumptions, it allows quickly estimating the permeability with satisfactory accuracy. Additional post-processing checking the validity of the resulting MIP was also performed: we ran comparisons between incremental *μ_MIP_* and differential *μ_Diff_* permeabilities (from Figure 9 results) in the saturated zone where they are supposed to be similar. These comparisons are available in Table 1. Figure 10 describes the method: reluctances *ℛ_1_* and *ℛ_2_* are associated with the ferrite core and the FeCo sheet, respectively. The magnetic contact between the ferrite core and the sheet is supposed ideal; thus, with a total absence of airgap. Then, by assuming *ℛ_2_>>*
*ℛ_1_*, this simplified approach leads to a linear relationship between the imaginary part of *Z″* and the relative permeability *μ_r_*:(3)$$\mu_{r}=\frac{Z^{\prime \prime }\cdot l_{2}}{\mu_{0}\cdot N^{2}\cdot A_{2}\cdot \omega }$$

Figure 11 gives the resulting *μ_r_(H_surf_)* for all Figure 9 tests. It can be noted that the curve shape is conserved compared to the impedance modulus *|Z|*, denoting negligible electrical losses in the device.

Table 2 comparisons confirm our expectations and the relatively similar permeabilities in the saturated range.

#### 3.2.4. From *μ_MIP_* to *B_a MIP_(H_surf_)* Hysteresis Cycles

The butterfly loop (Figure 11) is the MIP method conventional signature. In NDT, defining indicators read directly on this signature and plot them vs. the targeted properties (i.e., the property to be assessed, MIS, microstructural information, etc.) is classical. For MIP tests, those indicators include the maximum amplitude, the amplitude at the remanence point, the curve width at 50% of max(*μ_MIP_*), to name but a few [38]. In this study and for comparison purposes, the indicator definition has been postponed to a final step consisting of reconstructing a MIP hysteresis cycle.

Same indicators can be used for the classic and the MIP hysteresis cycles and comparing the stress influence on both these magnetic signatures is facilitated. The resulting induction field *B_a_*
_*MIP*_ is obtained from Equation (4):(4)$$B_{a\;MIP}=\int \mu_{0}\mu_{r\;MIP}.dH_{surf}$$

The integration process brings stability and eases the determination of the indicators. Figure 12 depicts the *B_a MIP_*(*H_surf_*) for all tests performed in Figure 9. 

#### 3.2.5. Directional *B_a MIP_(H_surf_)* Hysteresis Loop

In the following series of tests, five tensile stress levels and ten angle positions were implemented. The applied stress was limited to 250 MPa. For saturation reasons, the stress effect is especially weak beyond the [0–250] MPa range.

## 4. Discussion

*B_a_(H_surf_)* standard cycles are straightened by external tensile stress (Figure 7). Quite interestingly, *B_a MIP_(H_surf_)* loops show opposite behaviors (i.e., the resulting hysteresis loop is laying down, see Figure 13). Iron-Cobalt alloys are characterized by positive magnetostriction, resulting in a magnetic behavior softened by tensile stress. As an already soft material, tensile stress effects on *B_a_(H_surf_)* are limited. Oppositely, a much stronger influence can be detected on the *B_a MIP_(H_surf_)* cycles. This observation opens exciting perspectives and confirms MIP as an excellent alternative for estimating tensile stress stimuli.

Coercivity is supposed to be independent of the nature of the magnetic signature tested (*B_a_(H_surf_)*, Barkhausen noise, MIP, etc.). Therefore, comparing *H_c_(σ)* and *H_c MIP_(σ)* (at θ = 0 rad) can be considered a way to check the reliability of the MIP measurements. Figure 14 depicts this comparison in the *σ* = 0–250 MPa range (Figure 13), and as expected, coercivities of both charaterizations (*B_a_(H_surf_)* and *B_a MIP_(H_surf_)*) remain close as *σ* varies.

From its atomic origin to human-scale observation, magnetization in ferromagnetic steels depends on distinct mechanisms. Each mechanism is characterized by its geometrical scale, time constant, and sensitivity to the physical environment (magnetic, mechanical, thermal, etc.). These mechanisms overlap; thus, isolating and listing them is complex. Still, three categories can be established:Structure and kinetics of the magnetic domains (10^−4^–10^−6^ m):
Domain walls bulging (reversible, in the low excitation range);Irreversible domain wall motions (middle excitation range);Nucleation and annihilation (high range);Orientation and amplitude of atomic magnetic moments (10^−11^–10^−9^ m):
Magnetization rotation (high and very high magnetic excitation).Human scale mechanisms:
Macroscopic eddy currents

Such a combination makes impossible to isolate the effect of stress on a given mechanism by a unique observation of standard *B_a_(H_surf_)* hysteresis cycles. Oppositely, MIP experimental observation provides privileged access to domain walls bulging. In the recommendation of the literature [39,40,41], MIP is suggested to be run under an alternating magnetic field of amplitude half the coercivity, limiting the domain wall motions to reversible ones. Domain walls bulging is sensitive to the low amplitude of magnetic excitation. Scaling approaches and analogies encourage us to check its sensitivity in a low range of stress.

The directionality of domain wall bulging is an open question. Even if according to the manufacturer, FeCo laminations exhibit a not very pronounced crystallographic texture, every material is somehow anisotropic in its magnetic answer, and so it should be for MIP. 

Figure 13 confirms this statement by showing substantial variations over the tested angles. 

Five indicators directly read on the *q*-dependent *B_a MIP_*(*H_surf_*) cycles have been tested to identify the most adapted to a tensile stress estimation: Coercivity *H_c MIP;_*Remanence *B_r MIP;_**B_a MIP_*(*H_surf_*) Hysteresis area;*B_a MIP_* at *H_surf_* = 2 kA·m^−1;^*B_a MIP_* at *H_surf_* = 10 kA·m^−1.^

A refined analysis based on Pearson correlation factors *ρ* reveals *B_a MIP_* at *H_surf_ =* 10 kA·m^−1^ as the best indicator. Figure 15a depicts its variations vs. *σ* and for the ten sensor positions that have been tested. An astonishing 0.99 linear correlation was obtained for θ = π/2.

Even if read on the *B_a MIP_(H_surf_)* cycles, *H_c MIP_* depends on every magnetization mechanism. Oppositely, *B_a MIP_* at *H_surf_* = 10 kA·m^−1^ provides a cumulative reflection all along the magnetization cycle of the domain wall bulging mechanism.

Concerning the sensor orientation, there is no energetical reason for the domain wall to bulge differently in one direction or the other in a supposedly isotropic material. *σ* and *H_surf_* being parallel, on the first hand, if *θ* = 0 rad, the static and dynamic contributions are superimposed, creating an even stronger softening behavior, and accentuating the effect on the material and the behavior differences between the stress levels. Oppositely when *θ* = π/2 rad, the static and the dynamic contribution operate in quadratic directions, and the sensor measurement is less affected by the softening effect, so that the measure becomes quasi-linearly dependent on *σ*.

## 5. Conclusions

The directionality influence of MIP sensors has barely been studied before. In this work, we focus on observing the effect of a tensile stress *σ* on the magnetic response for several angles. We found that ideal conditions are reached when *H_surf DC_* and *H_surf AC_* exhibit quadratic directions.

*B_a_(H_surf_)* standard cycles are straightened by external tensile stress and show little changes. *B_a MIP_(H_surf_)* loops show opposite behaviors and significant variations (Figure 13), confirming MIP as an excellent way to estimate tensile stress stimuli.

Among *B_a_*
_*MIP*_*(H_surf_)* indicators, *H_c MIP_* depends on all the magnetization mechanisms. Its *σ* dependency is evident but not the most linear one. Other *B_a_*
_*MIP*_*(H_surf_)* indicators are more specific to the domain wall bulging mechanism and can be used to interpret the influence of *σ* on this mechanism. The best result in terms of linear correlation was obtained with *B_a_*
_*MIP*_ at *H_surf_* = 10 kA.m^−1^ and for *q* = π/2 rad (Pearson coefficient *ρ* ≈ 0.995).

Many possibilities can be listed in the follow-up associated with this study; this includes, in the short term, the influence of homogeneous compressive stress. Also, the proposed results should be confirmed on MIS of pre-characterized specimens (by X-ray diffraction, for instance). 

Eventually, a new type of materials (martensitic stainless steel, low carbon steel, etc.) could be tested. In an even longer-term vision, it would be interesting to check the capability of the directional MIP on other usual magnetic NDT targeted properties, including microstructural properties, aging, etc.

## Figures and Tables

**Figure 1 sensors-22-06296-f001:**
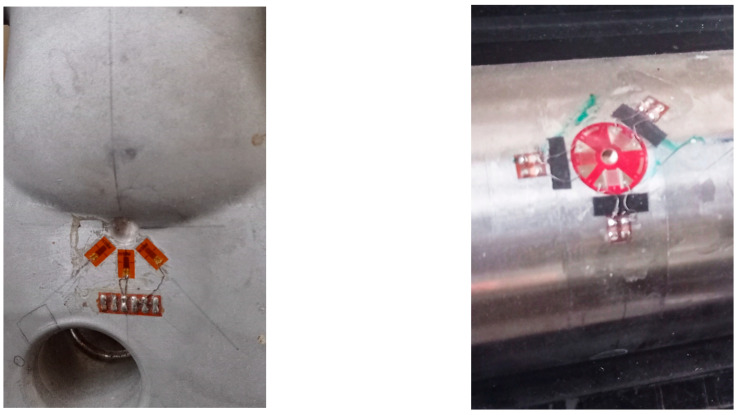
Illustration of internal stress evaluation by destructive hole-drilling tests (courtesy of CETIM, Senlis, France).

**Figure 2 sensors-22-06296-f002:**
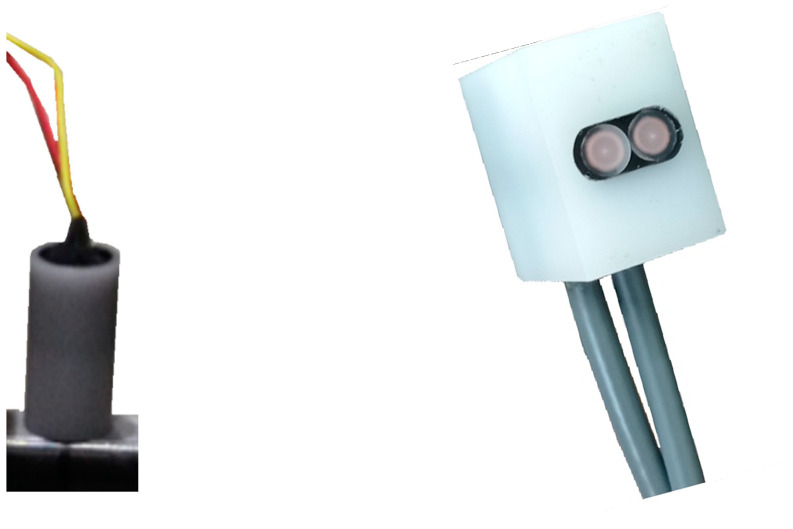
(**Left**): single pancake coil illustration; (**Right**): adjacent Transmitter/Receptor, TR-probes illustration.

**Figure 3 sensors-22-06296-f003:**
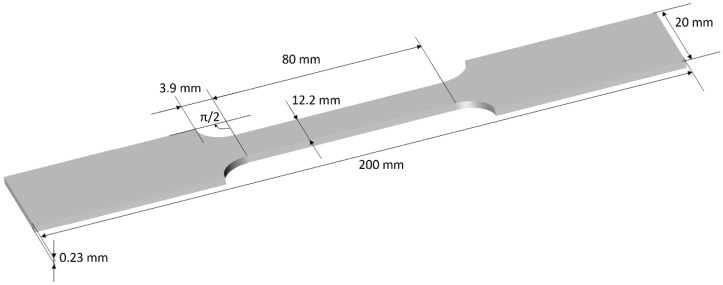
FeCo tensile stress specimen dimensions.

**Figure 4 sensors-22-06296-f004:**
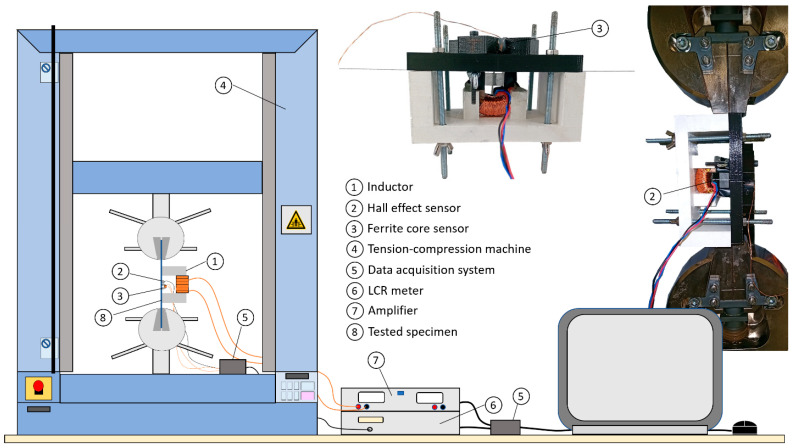
2D overview of the experimental setup.

**Figure 5 sensors-22-06296-f005:**
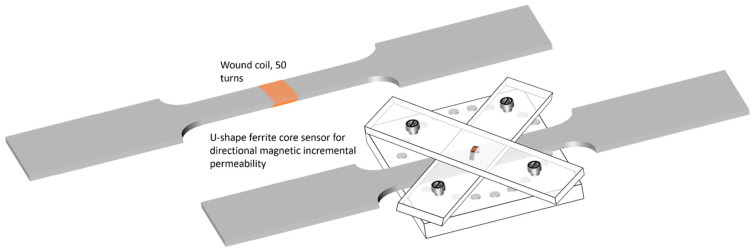
Illustration of the magnetic sensors used in this study.

**Figure 6 sensors-22-06296-f006:**
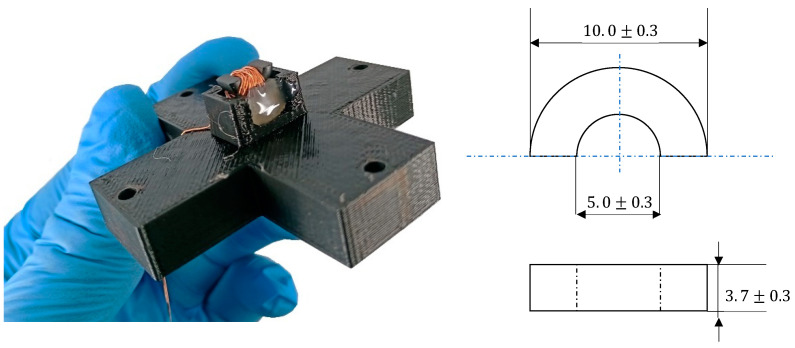
Photo and dimensions of the directional MIP sensor.

**Figure 7 sensors-22-06296-f007:**
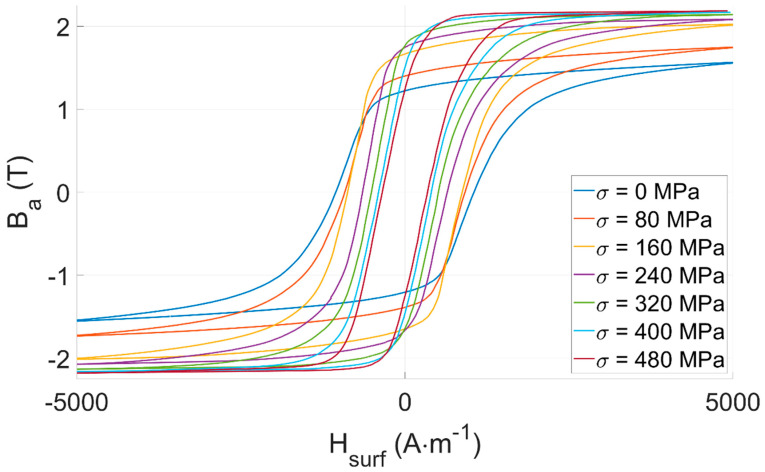
FeCo: experimental measurements for the tensile stress-dependent *B_a_*(*H_surf_*) hysteresis cycles.

**Figure 8 sensors-22-06296-f008:**
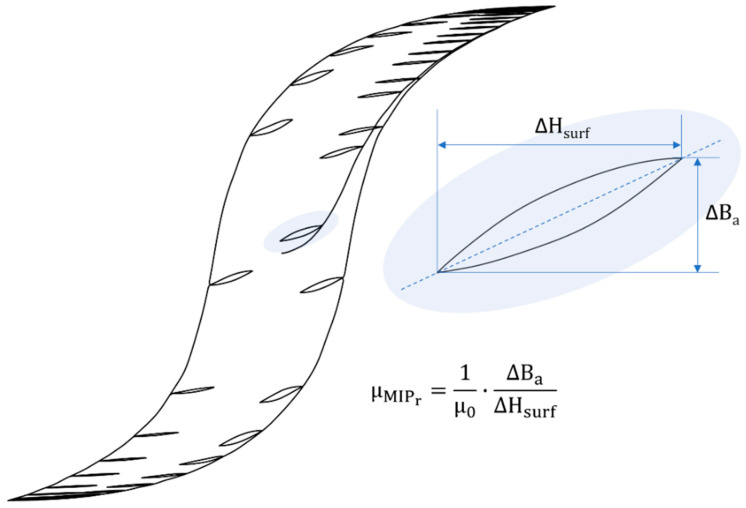
MIP illustration and equation [37].

**Figure 9 sensors-22-06296-f009:**
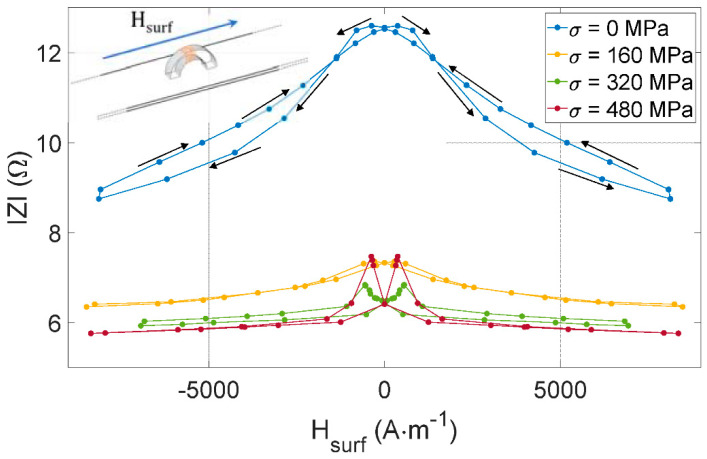
*IZI(H_surf_)* for different tensile stress levels and with the MIP sensor aligned with the length of the tested specimen as illustrated in the top left-hand side incrustation.

**Figure 10 sensors-22-06296-f010:**
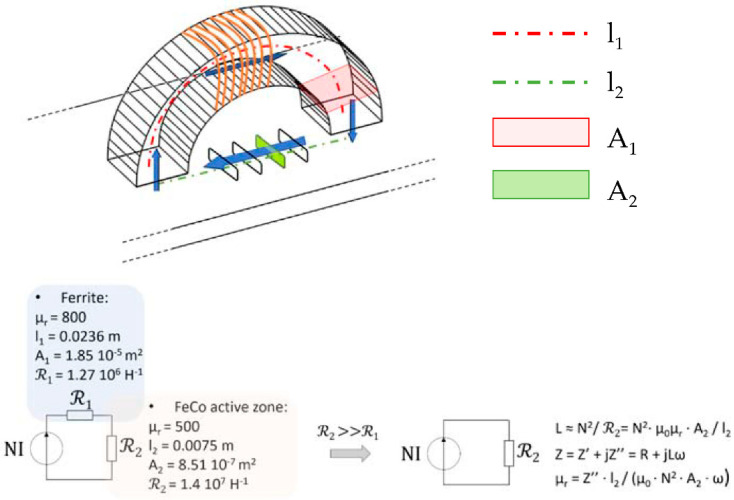
Simplified reluctance scheme for a linear relation between *Z*″ and *μ_r_*.

**Figure 11 sensors-22-06296-f011:**
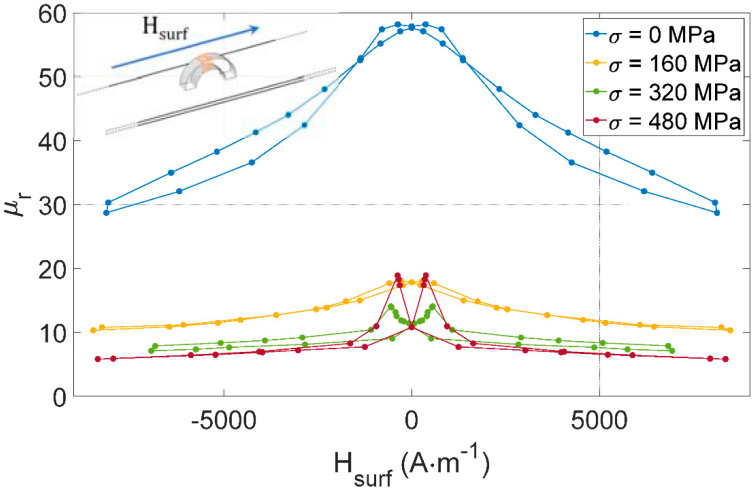
*μ_r_(H_surf_)* for different tensile stress levels and with the MIP sensor aligned with the length of the tested specimen as illustrated in Figure 9.

**Figure 12 sensors-22-06296-f012:**
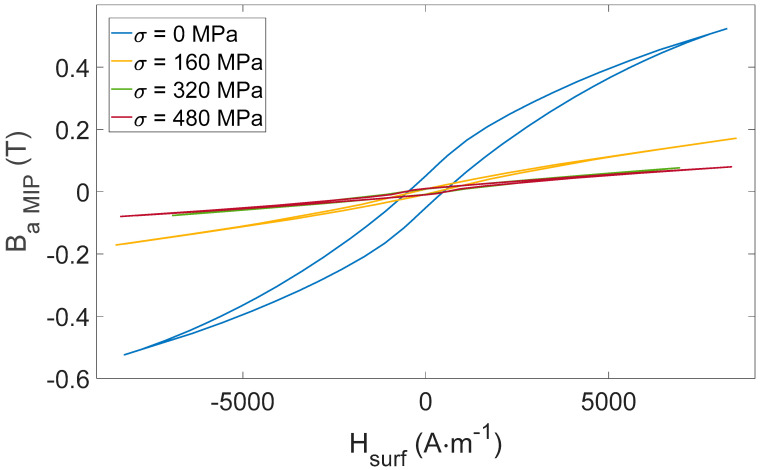
*B_a MIP_*(*H_surf_*) hysteresis cycles for all Figure 9 tests.

**Figure 13 sensors-22-06296-f013:**
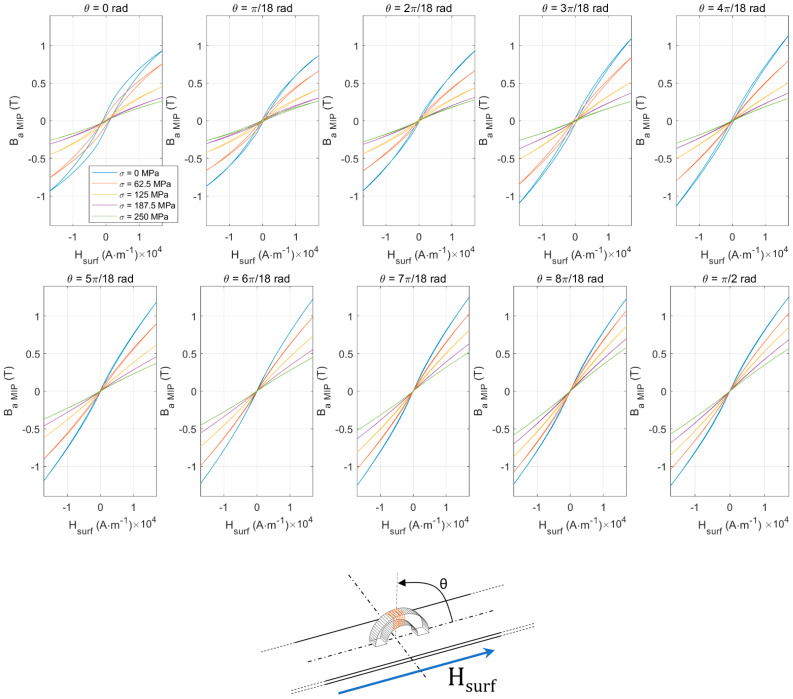
*B_a MIP_*(*H_surf_*) hysteresis cycles in the [0–π/2] angle range.

**Figure 14 sensors-22-06296-f014:**
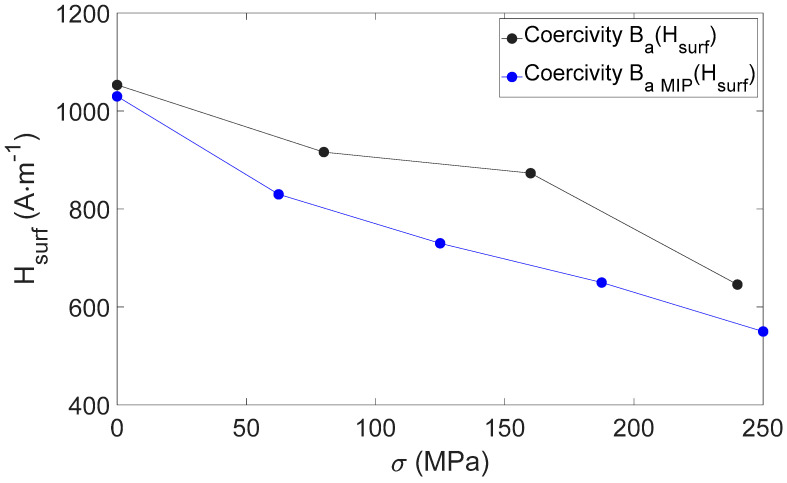
Coercivity vs. stress for *B_a_(H_surf_)* and *B_a MIP_(H_surf_)* hysteresis cycles.

**Figure 15 sensors-22-06296-f015:**
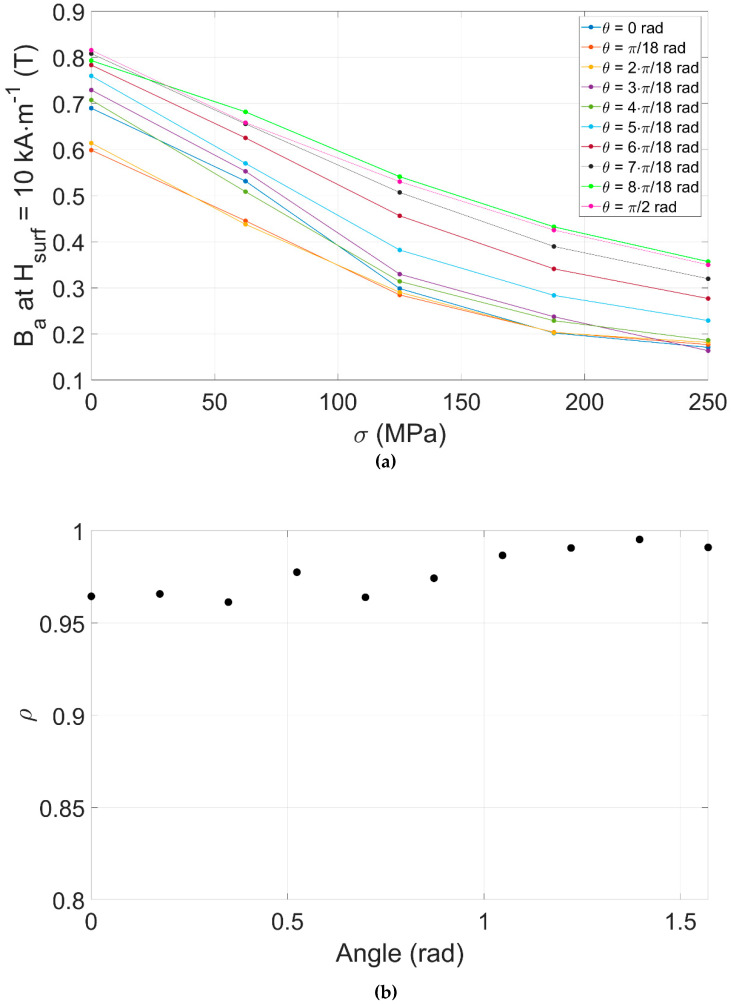
(**a**) *B_a_* at *H_surf_* = 10 kA·m^−1^ for θ in the [0, π/2] range. (**b**) Related Pearson correlation factors.

**Table 1 sensors-22-06296-t001:** Composition, physical and mechanical properties of the tested specimens.

Composition:					
**C (Mass %)**	**Si**	**Mn**	**Co**	**V**	**Fe**
<0.015	<0.1	<0.15	49	2	Bal.
Physical properties:					
**Density (g·cm^3^)**	**Elect. Res. (μΩ·cm)**	**Exp. Coef. (·K^−1^)**	**Therm. Cond. (W·cm^−1^ K^−1^)**	**Curie Temp. (°C)**	
8.12	40	9 × 10^−6^	0.3	950	
Mechanical properties:					
**Yield Strength (MPa)**	**Tens. Strength (MPa)**	**Young Mod. (GPa)**	**Hardness (HV10)**		
1000	1345	250	300		

**Table 2 sensors-22-06296-t002:** Comparisons between *μ_r MIP_* and *μ_r Diff_* in the high *H_surf_* amplitude range (saturated zone).

σ (MPa)	*H_surf_* (A·m^−1^)	*μ_r MIP_*	*μ_r Diff_*
–	4500	37.5	40
160	4500	12	17
320	4500	9	10
480	4500	7	9

## Data Availability

The study did not report any data.

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
