# Peer review of "Characterization of Tensile Stress-Dependent Directional Magnetic Incremental Permeability in Iron-Cobalt Magnetic Sheet: Towards Internal Stress Estimation through Non-Destructive Testing"

_sensors, 2022, doi:10.3390/s22166296_

Round 1
Reviewer 1 Report
The topic of this paper is interesting. And this paper can be accepted only after minor revision. My detailed comments are as follows.
(1) The section of ‘Introduction’ is too long. And authors should simplify ‘Introduction’.
(2) The keywords number is small, and also the first keywords ‘ directional magnetic incremental permeability’ needs to be changed.
(3) The conclusions are not concise, and need to be further rewritten.
Reviewer 2 Report
To improve the manuscript, please provide additional information and perform the following changes:
1) On page 2, lines 50-51, in the phrase “High power density gains are, for instance, expected with Fe-27%Co compound.” should be specified the expected values for power density gains.
2) On page 2, lines 56-57, in the phrase “There are many ways to reduce the magnetostrictive effect, including developing appropriate textures [4][6].” should be given some examples of the ways to reduce the magnetostrictive effect, and “appropriate textures” should be detailed.
3) On page 4, in 2.1. Description of the specimens should be specified the producer/supplier of the FeCo Iron-Cobalt laminations tested in this study. Referring to the composition should be clarified if it is in wt.%. The yield strength value of 1000 MPa should be clarified how was determined. To support the affirmations from lines 162-165 “FeCo laminations exhibit a relative isotropic magnetic behavior in the sheet plane in no-stress conditions. The crystallographic texture is not very pronounced, and the magnetocrystalline anisotropy is low” should be provided the corresponding magnetic and XRD analysis in the Methodology section and in the Results and discussion section. However, these comments should be moved from the Methodology section to the Results and discussion section.
4) On page 4, in Figure 3. FeCo tensile stress specimen dimensions should be drawn the length of the rectangular symmetrical parts and the radius, and their values should be specified.
5) On page 5, in subsection 2.2. Description of the experimental setup, referring to the tensile stress application should be specified the experimental conditions (test speed, distance between grips, initial load), and the number of tested specimens to check the reproducibility of the results.
6) The caption of Figure 5. Illustration of the magnetic sensors used in this study (page 6) should be on the same page as Figure 5 (on page 5).
7) On page 6, lines 241-242, referring to “Specific 3D-printed support (see Figure 5, 6) has been designed to hold the sensor…” should be clarified the methodology (the type and producer of 3D printer and experimental details) for obtaining the 3D-printed support and the results (material type and main properties) and should be provided comments if the 3D-printed support influenced the magnetic measurements.
8) The symbol for radian in SI is rad no rd. Therefore, throughout the manuscript and in the images related to rad should be changed rd with rad.
9) For all equations and Figure 8 (in the case it was taken from a literature report) should be added references.
10) In Figure 15. A, the Oy scale should be between 0.1 (to be visible on the scale) and 0.9.
11) In the Conclusions section should be provided quantitative data about the main results.
12) On page 15, line 632, referring to the comments “Eventually, a new type of materials could be tested…” should be given some examples of classes of materials because new materials are too generic expressed.
Round 2
Reviewer 2 Report
The authors performed in general a satisfactory revision, but a clarification for the starting materials (laminated FeCoV alloy) is needed. It is unclear why the authors did not use commercial laminated FeCoV alloys and performed their study on tensile-shape type specimens made by a non-identified European metallurgical group that intends to patent the laminated FeCoV alloy containing 49 wt.% Fe, 49 wt.% Co and 2 wt.% V. It is known that the properties of materials (metallic, ceramic, composites, magnetic, etc.) depend on the properties of the starting materials, as well as on the processing techniques and corresponding parameters. Therefore, the authors should define clearly the supplier and main characteristics of the starting materials or should use commercial laminated FeCoV alloys with well-defined suppliers and characteristics shown in technical sheets. In the first case, when novel or innovative alloys from R&D works are used, should be provided in the Methodology and Results and Discussion sections the methods and results for the investigation of density, chemical composition, and mechanical properties (e.g. yield strength, hardness) of the laminated FeCoV alloy used in this study to prove the claimed characteristics. Normally, the processing technique of the alloys should be also presented, but it is under the patenting process and is confidential. I think the identification of the supplier and characteristics of the starting materials (laminated FeCoV alloys) are very important in determining the properties of the developed sensors and in obtaining reproducible findings.
Round 3
Reviewer 2 Report
The manuscript is recommended for publishing in the Sensors journal since the authors performed a satisfactory revision.